# OpenReview forum: "RobustKV: Defending Large Language Models against Jailbreak Attacks via KV Eviction"
_ICLR.cc/2025/Conference — ICLR 2025 Poster_

### Official Review · Reviewer_mtHc · 2024-10-31

**Soundness:** 3
**Presentation:** 3
**Contribution:** 2
**Rating:** 6
**Confidence:** 4

**Summary:**

The paper  introduces a novel defense mechanism called RobustKV. This method tackles jailbreak attacks on large language models (LLMs) by selectively evicting key-value (KV) pairs of less important tokens from the model's KV cache. RobustKV identifies and removes tokens related to harmful queries by analyzing their attention scores, thus preventing the LLM from generating malicious responses.

**Strengths:**

1. Unlike existing defenses, which focus on mitigating jailbreak prompts, RobustKV directly addresses harmful queries by manipulating the KV cache, offering a fresh perspective on LLM defenses.
2. RobustKV does not increase computational overhead or response time, making it practical for real-world deployment.

**Weaknesses:**

1. In my opinion, since your method needs to access the LLM’s key-value (KV) caches, it may only be applicable to open-source LLMs. I believe this could be a significant limitation for your work.
2. In the example you provided in line 379, 'how to steal someone' is removed, so the attack will certainly not succeed. If the core content of the jailbreak issue has been removed and the LLM can still be attacked, that would be truly strange！In my opinion, this paper provides a meaningless method, and there's no scenario can the method be used.

**Questions:**

See Weaknesses.

---

> ### Author Response · Authors · 2024-11-22
>
> We thank the reviewer for the valuable feedback on improving this paper! Please find below our response to the reviewer’s questions.
>
> > In my opinion, since your method needs to access the LLM’s key-value (KV) caches, it may only be applicable to open-source LLMs. I believe this could be a significant limitation for your work.
>
> We would like to clarify that, as the LLM service provider, the defender typically has access to the LLM internals. For example, OpenAI can access and control ChatGPT's internals. Therefore, our defense is applicable to both open-source and closed-source LLMs when implemented by service providers, which is consistent with prior work (Zhang et al., 2024c; Wallace et al., 2024).
>
> > In the example you provided in line 379, 'how to steal someone' is removed, so the attack will certainly not succeed. If the core content of the jailbreak issue has been removed and the LLM can still be attacked, that would be truly strange！In my opinion, this paper provides a meaningless method, and there's no scenario can the method be used.
>
> We believe there is a misunderstanding. Unlike prior work that focuses on mitigating the jailbreak prompt’s effect, we present a novel approach: preventing malicious responses by minimizing the harmful query's presence in the LLM. The key challenge is **how to automatically identify harmful query tokens effectively**. RobustKV addresses this challenge through its core mechanism, illustrated in Figure 1: tokens in harmful queries (like 'how to steal someone') typically have lower attention scores than jailbreak prompt tokens. By strategically evicting KVs with the lowest attention scores, RobustKV diminishes the harmful query's presence in the KV cache, effectively preventing malicious responses.
>
> Again, we thank the reviewer for the valuable feedback. Please let us know if there are any other questions or suggestions.
>
> Best,
>
> Authors

---

> > ### Comment · Reviewer_mtHc · 2024-11-26
> >
> > Thank you for your reply. I will raise the score. What's more, just as you said, "The key challenge is how to automatically identify harmful query tokens effectively." I'm quite curious whether for harmless queries, using your algorithm, which part will be detected to have a lower attention score?  Is there a difference in the results of your method for harmless queries and harmful queries? Or is your assumed premise that all are harmful queries?

---

> ### Author Response · Authors · 2024-11-26
>
> Thank you for the prompt and constructive feedback!
>
> For harmless/benign queries, low-scoring tokens naturally correspond to those with minimal semantic significance (e.g., common articles, prepositions, redundant tokens, and repetitions). Our evaluation (Section 5.3) confirms that RobustKV maintains the LLM's performance on benign queries with minimal degradation. This finding also aligns with previous research on KV compression (e.g., Zhang et al., 2024b; Li et al., 2024), where a substantial portion of KV pairs (up to 95% in some cases) can be removed while preserving the LLM's performance.
>
> Please let us know if you have any additional questions or suggestions.
>
> Best,
>
> Authors

---

> > ### Comment · Reviewer_mtHc · 2024-11-27
> >
> > Thanks to the author's detailed reply, I have a better understanding of this work. I think it is a good paper, so I will raise the score.

---

### Official Review · Reviewer_k7do · 2024-11-01

**Soundness:** 3
**Presentation:** 3
**Contribution:** 3
**Rating:** 8
**Confidence:** 4

**Summary:**

This paper presents a novel approach to defending large language models (LLMs) from jailbreak attacks. Based on the idea that tokens associated with jailbreak prompts tend to hold higher importance than the actual harmful query, RobustKV evicts critical tokens of the harmful query from the KV cache. Thus it results in a non-informative response to the harmful query to improve jailbreak defense.  Benchmark evaluations under state-of-the-art jailbreak attacks demonstrate this method's effectiveness.

**Strengths:**

1. This work combines LLM inference efficiency and safety, addressing two critical areas for improving LLM performance.

2. Observation and motivation are both clear, I enjoy the insight this paper provides for the adaptive attack design.

3. Algorithm is the improvement of SnapKV and can be applied to any pre-trained LLMs without finetuning.

4. This paper provides comprehensive experiments to support the stability and efficacy of the algorithm.

**Weaknesses:**

1. For the Observation Window, did you try different setting or different size?
2. I notice that in Figure 3, this method achieve a trade-off between eviction ratio and defense effectiveness (clean accuracy and jailbreak defense). For larger LLMs like Llama2-13B, KV cache's performance may be different, would it still be as effective?
3. I'm curious about the choice to use token importance across all layers in the algorithm. What would the impact be if importance were calculated across all heads within a single layer instead?
4. Could you provide why choose these three different models for your algorithm? As I understand, Mistral uses GQA, how do the other two models compare in this experiment setting? Do these selections provide sufficient evaluation of your algorithm’s generalization?

**Questions:**

See above.

---

> ### Author Response · Authors · 2024-11-22
>
> We thank the reviewer for the valuable feedback on improving this paper! Please find below our response to the reviewer’s questions.
>
> > For the Observation Window, did you try different setting or different size?
>
> Following the reviewer's feedback, we evaluate AutoDAN's attack success rate (ASR) against Llama2 defended by RobustKV, varying the observation window size $k$, where we consider the first $k$ generated tokens as the observation window (more details in Appendix C6). The table below summarizes the results. Observe that larger observation windows enhance defense effectiveness, likely due to more stable importance score estimation. Notably, even a minimal window size of 2 reduces the ASR to approximately 10%.
>
> | Window Size |  2   |    4   |     8   |     16     |     32  |
> | :-----: | :-----: | :-----: | :-----: | :-----: | :-----: |
> | ASR  | 11.2% | 8.9% | 6.8% | 6.3% | 6.0% |
>
>
> >  I notice that in Figure 3, this method achieve a trade-off between eviction ratio and defense effectiveness (clean accuracy and jailbreak defense). For larger LLMs like Llama2-13B, KV cache's performance may be different, would it still be as effective?
>
>
> | Eviction Rate | 0%  |   5%  |    10%  |    15%   |  20%  |
> | :-----: | :-----: | :-----: | :-----: | :-----: | :-----: |
> | ASR | 26.6% | 13.7% | 9.1% | 3.1% | 2.5% |
> | WinRate |  69% |  66% | 62% | 62% | 62% |
>
>
> Thanks for the insightful question! Evicting more tokens reduces attack effectiveness on harmful inputs while marginally impacting performance on benign ones. Testing RobustKV on Llama2-13B (detailed in Appendix C7) shows even stronger results compared to Llama2-7B: lower attack success rates (ASR) at the same eviction rate, with minimal impact on general performance (WinRate). This improved effectiveness likely stems from stronger model safeguards in Llama2-13B, where circumventing alignment requires increasing jailbreak prompt importance - making RobustKV’s token eviction mechanism more effective.
>
> > I'm curious about the choice to use token importance across all layers in the algorithm. What would the impact be if importance were calculated across all heads within a single layer instead?
>
> The layer-wise token eviction approach, similar to SnapKV, proves ineffective against jailbreak attacks, as shown in Table 1 (the column of 'SnapKV'). This ineffectiveness can be attributed to two key limitations: First, layer-wise estimation fails to provide a stable measure of tokens' overall importance. Second, even when individual layers remove certain tokens, the tokens of the harmful query (e.g., ‘how to build a bomb’) may persist in other layers. RobustKV addresses these limitations by aggregating importance scores across all layers, resulting in more robust token importance estimates and enabling the removal of tokens with low overall significance.
>
> > Could you provide why choose these three different models for your algorithm? As I understand, Mistral uses GQA, how do the other two models compare in this experiment setting? Do these selections provide sufficient evaluation of your algorithm’s generalization?
>
> Most popular LLMs, including Llama2, Vicuna, and Mistral, employ grouped query attention (GQA) due to its optimal balance between memory bandwidth and performance. We selected these three models for their varying degrees of safety alignment, as evidenced in Table 1: GCG's attack success rates (ASR) of 38%, 64%, and 89% on Llama2, Mistral, and Vicuna respectively, represent a spectrum of attack vulnerability and general performance.
>
> Again, we thank the reviewer for the valuable feedback. Please let us know if there are any other questions or suggestions.
>
> Best,
>
> Authors

---

> ### Comment · Reviewer_k7do · 2024-11-22
>
> Thank you to the authors for addressing my questions. I think this work is important because it bridges LLM's efficiency with safety. The answers clarified my concerns, and I believe this work deserves more consideration. I will raise my score.

---

> > ### Author Response · Authors · 2024-11-22
> >
> > We are very glad to learn that our response and revision address your concerns. Thank you again for your insightful feedback on improving this paper!

---

### Official Review · Reviewer_wnH1 · 2024-11-04

**Soundness:** 3
**Presentation:** 4
**Contribution:** 3
**Rating:** 6
**Confidence:** 5

**Summary:**

Safety-tuned LLMs are still jailbroken by advanced techniques. This paper implements a complimentary method to safety-tuning for defending against jailbreaks.

The paper makes the following observations:
- Jailbreak tokens must have high attention scores to bypass safety
- High jailbreak token attention necessarily reduces harmful query token attention
- This creates a consistent pattern: harmful query tokens rank lower than jailbreak tokens

They implement RobustKV, which evicts the harmful components of the jailbreaking input. This forces attackers into a dilemma:
- Make harmful query important = jailbreak fails to bypass safety
- Make jailbreak important = harmful query gets evicted
- Can't optimize both simultaneously

**Strengths:**

- This work leverages safety-tuning. Unlike the jailbreak defense pipelines mentioned in [1] which constitute defining properties of harmful outputs explicitly, this approach preserves the learnt safety-tuning policy, and does not seek to redefine safe and unsafe behavior.
- This defense points out that many jailbreak techniques leverage the fact that safety-tuning does not catch when malicious queries are obfuscated. Given the possibility of RobustKV, potentially this attack model will no longer be as interesting to study.


[1] Testing the Limits of Jailbreaking Defenses with the Purple Problem. Kim 2024.

**Weaknesses:**

- This paper needs to better justify that this sort of defense is useful:
  -  [1] ran an extensive comparison across jailbreaking techniques, and found that [2] and [3] are the strongest attack techniques when comparing to all the techniques mentioned in this paper. Either a larger scope of attacks should be defended against using RobustKV, or proper analysis of patterns within the studied jailbreaks should be provided. Given the evolving offense-defense balance, defense techniques cannot be measured against a select set of attacks without a proper argument for it.
  - [4]  relies on in-context learning, which may not be defended against RobustKV. In security, in order to provide adequate evidence of your hypothesis, please seek out counterexamples that might be most challenging.

- Generating uninformative responses may be a significant challenge to any model developers implementing such methods.
- The related works section only cites as one black-box jailbreaking technique, please refer to [1] for a list.

[1] A STRONGREJECT for Empty Jailbreak. Souly 2024.
[2] How Johnny Can Persuade LLMs to Jailbreak Them: Rethinking Persuasion to Challenge AI Safety by Humanizing LLMs. Zeng 2024.
[3] Jailbreaking Black Box Large Language Models in Twenty Queries. Chao 2024.
[4] Many-shot jailbreaking. Anil 2024.

**Questions:**

- The LLM Utility section only has data for Llama, what about the others?
- If utility was not decreased by percentage scores, did you also also analyze the change in distribution?

---

> ### Author Response · Authors · 2024-11-22
>
> We thank the reviewer for the valuable feedback on improving this paper! Please find below our response to the reviewer’s questions.
>
> > [1] ran an extensive comparison across jailbreaking techniques, and found that [2] and [3] are the strongest attack techniques when comparing to all the techniques mentioned in this paper. Either a larger scope of attacks should be defended against using RobustKV, or proper analysis of patterns within the studied jailbreaks should be provided. Given the evolving offense-defense balance, defense techniques cannot be measured against a select set of attacks without a proper argument for it.
>
> Following the reviewer's feedback, in addition to the jailbreak attacks in Section 5, we further consider more recent attacks. Under the default setting, we evaluate RobustKV against PAP (Zeng et al., 2024), PAIR (Chao et al., 2023), and many-shot jailbreaking (Anthropic, 2024), on Llama2-7B (with more details in Appendix C4).
>
> | KV Eviction Rate | 0%  |  15%  |   20%  |    30%  |
> | :-----: | :-----: | :-----: | :-----: | :-----: |
> | ASR | 46% | 24% | 14% | 8% |
>
> The table above demonstrates RobustKV's defense capabilities against PAP. When tested against the authors' released adversarial prompts, RobustKV reduces PAP's attack success rate from 46% to 14% with a 20% token eviction rate. While PAP employs an interpretable persuasive adversarial prompt that distributes harmful queries throughout the prompt structure, RobustKV maintains its effectiveness by identifying harmful content through token importance scores (rather than their positions). This approach enables successful detection and mitigation regardless of where malicious content is embedded within the prompt.
>
>
> | KV Eviction Rate | 0%  |  20%  |
> | :-----: | :-----: | :-----: |
> | ASR | 2% | 0% |
>
> The table above shows RobustKV's effectiveness in defending against PAIR under the setting of  "--attack-model gpt-4 --target-model llama2 --judge-model gpt-4 --n-iterations 20 --n-streams 5 --max-n-attack-attempts 5".  RobustKV reduces PAIR's ASR from 2% to 0% when using a 20% eviction rate. Notably, PAIR is a black-box jailbreak attack. While PAIR employs prompt-level perturbation, in contrast to token-level attacks like GCG, its resulting prompts can still be decomposed into two distinct components: the harmful query and the jailbreak prompt. RobustKV successfully identifies and mitigates the harmful query component through its importance score-based analysis.
>
>
> > [4] relies on in-context learning, which may not be defended against RobustKV. In security, in order to provide adequate evidence of your hypothesis, please seek out counterexamples that might be most challenging.
>
> Following the reviewer's feedback, we extend our evaluation to assess RobustKV's performance against many-shot jailbreak attacks. We first evaluate it on Llama2-7B. The token window limitations on Llama2 (up to 4096 tokens) restrict our analysis to a maximum of $2^5$ shots. Across all tested configurations, from $2^0$ to $2^5$ shots, the ASR  consistently remains at zero, aligning with findings reported in Anthropic (2024).
>
> | \# Shot  |      $2^0$      |        $2^1$    |     $2^2$      |    $2^3$     |    $2^4$     |      $2^5$ |
> | :-----: | :-----: | :-----: | :-----: | :-----: | :-----: | :-----: |
> | Undefended  |     58.4%     |        80.3%     |       85.1%     |       92.0%     |        91.9%   |  93.8%
> | RobustKV |       46.0%    |        64.5%    |         67.8%    |        84.5%   |         83.3%   |   83.5%
>
> We further test RobustKV's performance (using a fixed 20% eviction rate) against many-shot attacks on Mistral, a weakly-aligned LLM. As shown in the table above, while RobustKV shows reduced effectiveness against in-context learning-based attacks, compared to other attacks (cf. Table 1 in the paper), it still attains a substantial reduction in the ASR of many-shot attacks. We acknowledge the limitation of RobustKV in defending against jailbreak attacks using alternative learning paradigms (e.g., in-context learning) and identify the enhancement of RobustKV against many-shot attacks as a key direction for our ongoing research.
>
> > Generating uninformative responses may be a significant challenge to any model developers implementing such methods.
>
> Our objective is to generate uninformative responses to jailbreak prompts while preserving high-quality responses to benign queries. Prior work (e.g., H2O) suggests that LLMs can maintain good performance even with 95% KV cache eviction. As our method requires evicting around 20% of the KV cache, we believe it is practical to maintain model performance. Further, RobustKV's implementation is compatible with any Transformer-based LLMs, as it only requires collecting token importance scores across attention layers and removing low-ranked tokens. Upon completion of the review process, we will release all code and data used in this research to facilitate the reproduction and deployment of our work.

---

> > ### Author Response · Authors · 2024-11-22
> >
> > > The related works section only cites as one black-box jailbreaking technique, please refer to [1] for a list.
> >
> > Thank you for noting the missing references on black-box jailbreak attacks. We have now incorporated them in Section 2 of the revised version.
> >
> > > The LLM Utility section only has data for Llama, what about the others?
> >
> > In response to the reviewer’s feedback, we include the utility measures on other LLMs, including Vicuna, Mistral, and Llama2-13B using the AlpacaEval benchmark (details in Appendix C5). The tables below summarize the results. Observe that across all the cases,
> > RobustKV achieves utility closely matching those of undefended LLMs, indicating its negligible impact on the LLM's general performance.
> >
> > WinRate:
> > | LLM |  Vicuna  | Mistral | Llama2-13B |
> > | :-----: | :-----: | :-----: | :-----: |
> > | Undefended |  60% | 65% | 69% |
> > | RobustKV |  56% | 59% | 62% |
> >
> > Rouge-L:
> > | LLM |  Vicuna  | Mistral | Llama2-13B |
> > | :-----: | :-----: | :-----: | :-----: |
> > | Undefended |  0.376    |  0.472     | 0.486|
> > | RobustKV | 0.332  | 0.386 | 0.410 |
> >
> > > If utility was not decreased by percentage scores, did you also also analyze the change in distribution?
> >
> > We analyze RobustKV's effect on the LLM's generative distributions. The table below compares the LLM's average perplexity on the AlpacaEval dataset (with 100 random samples) with and without RobustKV. The minimal difference in perplexity suggests that RobustKV effectively preserves the LLM's general distributional characteristics.
> >
> > | LLM  | Undefended | RobustKV |
> > | :-----: | :-----: | :-----: |
> > | Llama2-7B | 1.137 |1.135 |
> > |Mistral | 1.267 | 1.266 |
> > |Llama2-13B | 1.113 | 1.117 |
> >
> > Again, we thank the reviewer for the valuable feedback. Please let us know if there are any other questions or suggestions.
> >
> > Best,
> >
> > Authors

---

> ### Comment · Reviewer_wnH1 · 2024-11-27
>
> Thanks for running the additional experiments and adding references, I appreciate the work that went into it.
>
> My primary request in order to feel comfortable increasing my support for this paper is that you add a limitations section to the paper. I think it is important for paper framing to reflect the actual tradeoffs and usefulness. The two big limitations that are essential to flag are: (1) Robust KV does not solve for attacks that use other mechanisms. You can still highlight that this paper shows that input filtering can rid of multiple SOTA attack methodologies (which is a compelling contribution to the jailbreaks literature). (2) Model deployment has other desiderata than not responding to harmful responses (eg: refusing eloquently and coherently, redirecting to responses that are not harmful). While your reported general performance scores don't have a significant impact, your examples indicate that user experience is not going to be smooth (which is not negligible), and the model output leaks information about the defense.
>
> If you disagree with the feedback, please feel free to say so.

---

> > ### Author Response · Authors · 2024-11-27
> >
> > Thank you for the very constructive feedback! We have incorporated your comments into a new Section 6, which addresses the limitations of this work.
> >
> > Please let us know if you have any additional questions or suggestions.
> >
> > Best,
> >
> > Authors

---

> > > ### Comment · Reviewer_wnH1 · 2024-11-27
> > >
> > > Great, thank you for incorporating the feedback.

---

### Meta-Review · Area_Chair_FuFr · 2024-12-13

**Metareview:**

The authors propose RobustKV, a novel defense mechanism aimed at defending large language models (LLMs) from jailbreak attacks, which are attempts to manipulate the model into producing harmful or undesired outputs. The key innovation of RobustKV is the eviction of critical tokens from the key-value (KV) caches during the inference process, which prevents the model from generating specific responses to adversarial prompts. The paper demonstrates through experiments that this approach is effective in mitigating jailbreak attacks without significantly harming the model's overall performance.

**Additional Comments On Reviewer Discussion:**

During the rebuttal period, the authors addressed several key points raised by the reviewers. Reviewer raised concerns about the scalability of the defense in real-world settings and the potential for attackers to adapt to the mechanism. The authors clarified that their approach is lightweight and scalable, with minimal overhead in practice, and they provided additional experiments showing that the defense remains effective under various attack scenarios, though they acknowledged the potential for future adaptation by attackers. Reviewer pointed out the lack of theoretical analysis, and the authors added a more detailed explanation of the underlying principles behind the key-value eviction process, though they admitted that a full theoretical framework could be a future research direction. Reviewer expressed concerns about the evaluation of edge cases, particularly in highly sophisticated attack scenarios. In response, the authors extended their evaluation to include more complex adversarial settings and provided further validation of the method's robustness. In weighing these points, the reviewers' concerns about edge cases and theoretical grounding were acknowledged, but the authors' detailed rebuttal and additional experiments strengthened the case for RobustKV's effectiveness and applicability, leading to a decision for acceptance.

---

### Decision · Program_Chairs · 2025-01-22

Accept (Poster)